# CoReDiT: Spatial Coherence-Guided Token Pruning and Reconstruction for Efficient Diffusion Transformers

## Abstract

Diffusion Transformers (DiTs) have achieved remarkable results in image and video generation, but their high computational cost limits scalability and deployment. We introduce CoReDiT, a general-purpose token pruning framework across vision tasks tailored for DiTs. CoReDiT leverages spatial coherence to estimate token redundancy within local latent grids and selectively skips high-coherence tokens during self-attention. To preserve visual fidelity, we reconstruct the skipped token outputs through similarity-weighted aggregation from spatially neighboring retained tokens that have participated in self-attention computation. In addition, we propose a progressive pruning schedule that dynamically adapts pruning ratios across transformer blocks and denoising steps based on redundancy statistics. Applied to state-of-the-art diffusion backbones such as PixArt-$\alpha$ and MagicDrive-V2, CoReDiT achieves up to **55%** reduction in self-attention FLOPs and latency speedups of **1.33**$\times$ on cloud GPUs and **1.72**$\times$ on mobile NPUs, while maintaining high visual quality. Moreover, CoReDiT enables significantly higher resolution generation on mobile devices. Our results demonstrate that spatial coherence is a powerful signal for structured pruning in diffusion transformers.

## 1 Introduction

Diffusion models learn to synthesize data through an iterative denoising process; this paradigm has achieved state-of-the-art results across diverse applications, such as conditional (class labels, text, edge, depth maps) image generation, image enhancement (inpainting, super-resolution) and video synthesis. Building on this success, Diffusion Transformers (DiTs) [21] replace convolutional U-Nets with attention-based backbones that effectively model long-range dependencies between tokenized patches; this architecture can demonstrate remarkable scalability with increased model size and facilitate incorporating conditioning from various modalities.

However, these benefits come with demanding computational cost. The complexity of transformer attention scales quadratically with the number of tokens, which poses substantial challenges as image resolutions or number of video frames increase, especially on mobile platforms with tight memory budgets and constrained computational capacity. In fact, much of attention computation is spent on tokens from visually redundant, low-saliency regions (e.g., uniform backgrounds and smooth textures). Motivated by this observation, a line of works proposes to prune the token sequence, i.e., selecting only the most informative tokens to participate in attention, such as saliency signals (e.g., attention scores [13, 27]), similarity (e.g., ToME [1]) and learnable predictors (e.g., DynamicViT [23], DiffCR [32]). However, several challenges remain: (1) localizing low-saliency tokens efficiently and effectively, (2) preserving visual semantics for the tokens that do not participate in attention, and (3) determining a pruning schedule that adapts across layers and denoising timesteps.

**Contributions:** To address these challenges, we propose CoReDiT, a token pruning framework for pre-trained DiTs, which incorporates the following components. (1) *Spatial coherence-based selector*: Redundant tokens tend to be highly similar to their spatial neighbors. We exploit this by partitioning the token lattice into small, non-overlapping grids and computing a spatial coherence score for each token, i.e., its average feature similarity to tokens in the same grid. Accordingly, tokens with the highest coherence are redundant and bypass attention. This mechanism is hardware-friendly and introduces only minimal overhead. (2) *Coherence-based reconstruction*: Simply skipping to-

kens can harm locality and result in artifacts. Therefore, we reconstruct skipped tokens from their retained neighbors via similarity-weighted aggregation. This content-aware interpolation preserves visual semantics and avoids artifacts or texture loss that arise from zeroing or naive forwarding. (3) *Progressive, block-adaptive pruning*: Pruning tolerance is uneven: different blocks carry varying semantic significance, and late diffusion steps are less tolerant than early ones. Inspired by this, we adopt a progressive, block-adaptive schedule during fine-tuning. Every fixed number of steps, we compute a redundancy score per block as the sum of the top-$\Delta k$ coherence scores, and greedily increase the pruning ratio for the most redundant block (with timestep weighting to protect late steps). This concentrates pruning on blocks where the model is more resilient, and introduces capacity reduction gradually to ensure stability.

Combining these components results in an efficient pruning pipeline that preserves high-fidelity outputs while reducing attention computation. Applied to text-to-image and video generation, including the autonomous driving model MagicDrive-V2, CoReDiT enables higher-resolution synthesis and maintains stable conditional alignment. Overall, it achieves up to 55% savings in self-attention FLOPs and latency speedups of $1.33\times$ on cloud GPUs and $1.72\times$ on mobile devices.

## 2 RELATED WORK

**Diffusion Models.** Diffusion models have recently achieved remarkable results in various generative applications across image, video, and 3D domains. Early diffusion models adopt a U-Net [25] backbone, which couples multi-scale convolutional encoders/decoders with skip connections to propagate fine detail while denoising across noise levels [12]. To reduce computational cost and enable high-resolution synthesis, diffusion models often operate in a compressed latent space: an autoencoder maps pixels to a low-dimensional latent grid where denoising occurs, then decodes back to pixel space [24]. More recently, transformer-based backbones (DiTs) [21] have emerged as alternatives to U-Net. These models patchify latents and use attention to capture long-range dependencies with a uniform block structure, demonstrating strong scalability for higher resolutions, longer videos, and multi-modal conditioning.

**Efficient Diffusion Transformer.** Due to the quadratic cost in memory and computation, a substantial body of work has focused on designing efficient DiT models: (1) *Token pruning*. Prior Works reduce the effective token either by merging redundant latents in a training-free way (e.g., ToMe variants [2]) or by learning importance scores/keep-ratios that vary by layer and timestep [32]. (2) *Weight-pruning and architecture editing*. Beyond tokens, structured and unstructured pruning remove channels, heads, or even full DiT blocks, with brief recovery fine-tuning or lightweight calibration to retain quality [9]. Other work (e.g., grafting [5]) edits the architectures of pretrained DiTs to explore more efficient backbones under small compute budgets; these approaches compose naturally with token sparsity. (3) *Caching and reuse across timesteps*. Feature-reuse methods [30] exploit the smooth evolution of hidden states across denoising steps, reusing block/layer activations with policies that decide when to refresh. Later work [19] brings this to DiTs with token-wise or layer-wise selection and adds forecasting/correction to mitigate drift.

**Token Pruning.** Token pruning is an inference-time acceleration strategy that skips computation on less important tokens (patch embeddings) in a transformer. Existing token pruning work can be categorized into saliency signals (e.g., attention scores [13, 27]), similarity (e.g., ToME [1]) and learnable predictors (e.g., DynamicViT [23]). Existing token pruning methods have achieved promising results in ViTs for simple tasks (e.g., classification and object detection) [16], where the objective primarily focus on key tokens and tolerate token dropping. In DiT, by contrast, naively skipping tokens can result in visual discontinuities in the generated images (e.g., inconsistent textures, distorted boundaries). Recent DiT-specific work dynamically modulates token density across layers/timesteps (e.g., FlexDiT [6], DiffCR [32]).

## 3 PROPOSED APPROACH: COREDIT

Fig. 1 summarizes the workflow of CoReDiT: In each transformer block, we partition the input tokens (i.e., patch embeddings) $X$ into a retained set $X_R$ and a skipped set $X_S$ according to a novel spatial coherence score of each token; only $X_R$ participates in multi-head self-attention, producing $Y_R = \mathrm{MHSA}(X_R)$ (Section 3.1). For skipped tokens $X_S$, we synthesize their attention outputs $Y_S$ using the attention outputs of the retained tokens $Y_R$ from neighboring spatial positions, guided

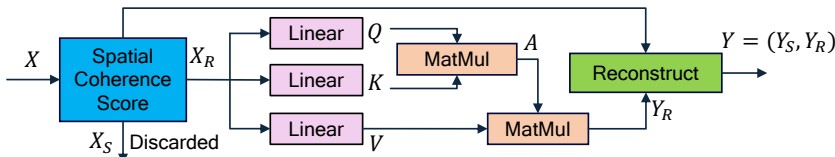

Figure 1: Workflow of CoReDiT. Token pruning is applied within each transformer block. The input tokens $X$ are split into a retained $X_R$ and skipped $X_S$ sets using the proposed spatial coherence score, which measures token similarity within local neighborhood (Section 3.1.2). Only the retained tokens $X_R$ participate in self-attention, improving effciency by skipping $X_S$. Skipped tokens are later reconstructed using the self-attention output $Y_R$ and the spatial coherence score (Section 3.2).

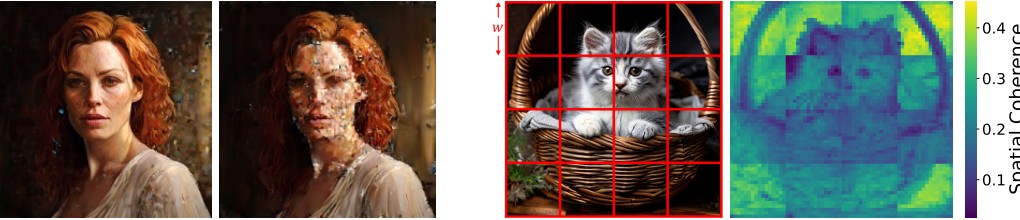

(a) Token pruning based on attention scores: removing tokens with the lowest scores (left) vs the highest scores (right)

(b) Visualization of the proposed spatial coherence score. Left: original image with grid partition overlay. Right: computed spatial coherence score.

Figure 2: Token selection motivation and visualization.

by token similarity (Section 3.2). To effectively determine the pruning schedule, we propose a progressive pruning strategy: during fine-tuning of a pretrained DiT, the pruning ratio for each block gradually increases based on the estimated token redundancy (Section 3.3).

## 3.1 TOKEN SELECTION

### 3.1.1 MOTIVATION

We begin with a commonly used criterion for token pruning: attention score-based importance [13, 27]. Let $A = \text{softmax}(\frac{QK^T}{\sqrt{d_k}})$ denote the attention matrix, where the sum of column in $A$ measures the total attention token received from all other tokens. Accordingly, we conduct experiments of pruning tokens with highest v.s. lowest attention scores for a DiT, as shown in Fig. 2a. As can be seen, pruning tokens with high attention score severely impacts the visual semantics, since these tokens concentrate on salient image regions (e.g., edges, textures, object boundaries); in contrast, pruning tokens with low attention score results in smaller quality degradation because they primarily cover low-salience background areas and contribute less to the transformation of salient tokens. However, computing full attention score $A$ involves quadratic complexity in time and memory, which diminishes the efficiency gains from token pruning.

A key observation is that low-saliency tokens (e.g., background) exhibit *strong local redundancy*: their patch embeddings are highly similar to spatially nearby tokens. Motivated by this, we propose an efficient token selection mechanism that leverages the similarity between a token and its neighboring tokens within each local region, which, as we shall see, is both effective and incurs minimal computation overhead.

### 3.1.2 SPATIAL COHERENCE

Given $N$ tokens (patch embeddings) $X = \{x_i\}_{i=1}^{N}$ reshaped to a 2D spatial layout $H \times W$, we partition tokens into non-overlapping grids of shape $w \times w$, as shown in Figure 2b. For a token $x_i$, we define its spatial coherence (SC) as the average similarity to all tokens in the same grid:

$$\text{SC}(x_i) := \frac{1}{|\mathcal{G}(i)|} \sum_{j \in \mathcal{G}(i)} \text{sim}(x_i, x_j). \tag{1}$$

where $\mathcal{G}(i)$ denotes the set of token indices that belong to the same grid as token $i$. Specifically, we adopt cosine similarity $\text{sim}(x_i, x_j) = \frac{x_i^T x_j}{||x_i||_2 ||x_j||_2}$ due to its computational efficiency and strong

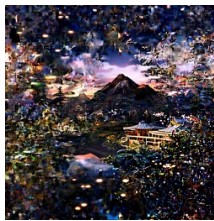 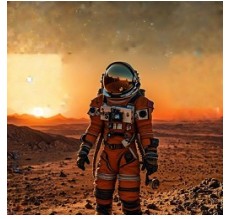 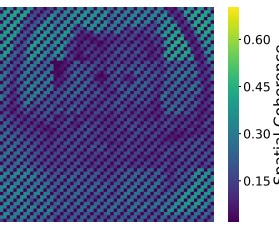

(a) Simply skipping tokens     (b) Visual artifacts on border     (c) $m$-stride token retention

Figure 3: Issues and proposed improvements. (a) Naively skipping tokens causes inconsistent textures, which can be mitigated by token reconstruction. (b) Using a fixed grid size during partitioning causes visual artifacts along grid borders; alternating grid sizes across transformer blocks help reduce these artifacts. (c) $m$-stride token retention ensures that at least one token is retained for every $m$ tokens during pruning.

empirical performance in prior work [1]. We average within each grid so that scores are comparable across transformer blocks with different grid size $w$, which is useful for the progressive pruning schedule in Section 3.3. Note that the reason why partition tokens into grids is to capture the local spatial coherence among the image patches.

Naively, computing all within-grid pairwise similarities incurs $O(w^2 N)$ complexity. Let $\hat{x}_i = \frac{x_i}{||x_i||_2}$ denote the normalized token embedding, and $\hat{g}_i = \frac{1}{|\mathcal{G}(i)|} \sum_{j \in \mathcal{G}(i)} \hat{x}_j$ denote the the mean normalized embeddings within the grid of token $x_i$. Then we can compute the spatial coherence score efficiently

$$\text{SC}(x_i) = \frac{1}{|\mathcal{G}(i)|} \sum_{j \in \mathcal{G}(i)} \hat{x}_i{}^T \hat{x}_j = \frac{\hat{x}_i{}^T}{|\mathcal{G}(i)|} \sum_{j \in \mathcal{G}(i)} \hat{x}_j = \hat{x}_i{}^T \hat{g}_i. \tag{2}$$

Essentially, the spatial coherence is an inner product between a token's normalized embedding and the mean normalized embedding of its grid. This reduces the complexity of score computation to $O(N)$, regardless of grid size $w$, since $g_i$ can be computed once per grid and reused for all tokens in that grid.

Figure 2b visualizes the spatial coherence score on a pretrained PixArt-$\alpha$-1024 model. For $64 \times 64$ input tokens, we choose grid size 16 or 9 to partition the tokens into $4 \times 4$ or $7 \times 7$ grids. As can be seen, high-coherence scores concentrate in low-saliency regions that are locally redundant, while edges and textured structures exhibit low coherence. Accordingly, our selection mechanism skips tokens with highest spatial coherence score in a transformer block to preserve visual quality: $X_S = \{x \in X \mid \text{token } x \text{ has the top-}K \text{ spatial coherence values}\}$. Note that our proposed spatial coherence is not only used for token selection, but also utilized to update skipped tokens and guide our progressive pruning, as we discuss in the following.

## 3.2 TOKEN RECONSTRUCTION

### 3.2.1 MOTIVATION

Token pruning has achieved promising results in ViTs for simple tasks such as classification and object detection [16], where the objectives primarily focus on key tokens and can tolerate token dropping. In DiT, in contrast, naively skipping tokens can result in visual discontinuities in the generated images (e.g., inconsistent textures, distorted boundaries), as shown in Fig. 3a. Consequently, to preserve visual semantics of the generated images, we propose to reconstruct the transformed version of the skipped tokens based on the retained tokens that have gone through self-attention in the local neighborhood, by leveraging their correlation relationship. In particular, since the proposed selection mechanism is inclined to skip redundant tokens with high coherence, these tokens can be more effectively reconstructed.

### 3.2.2 COHERENCE-BASED RECONSTRUCTION

To compute the block transformation $Y$ from input tokens $X$, we identify a skipped set $X_S$ and forward only the retained tokens $X_R = X - X_S$ through multi-head self-attention to obtain $Y_R = \text{MHSA}(X_R)$. To preserve spatial continuity without performing attention on $X_S$, we synthesize the

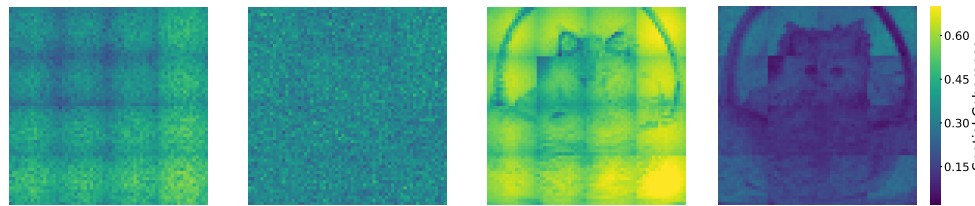

(a) First step, first block  (b) First step, last block  (c) Last step, first block  (d) Last step, last block

Figure 4: Motivation for coherence-based pruning ratio across blocks and time steps (a comprehensive version is provided in Fig. 10). (a) Early timestep in the first block: structured patterns dominated by noise and global semantics. (b) Early timestep in the last block: reduced structure, reflecting weaker locality. (c) Late timestep in the first block: refined spatial details with clearer object-level coherence. (d) Late timestep in the last block: partial refinement with moderate spatial structure. These observations suggest that both timestep and block position influence locality, motivating the use of adaptive and dynamic pruning ratios

missing transformations $Y_S$ by similarity-weighted aggregation over nearby retained tokens:

$$\forall x_i \in X_S, \quad y_i = \frac{\sum_{j\in\mathcal{S}(i)} \text{sim}(x_i, x_j) \cdot y_j}{\sum_{k\in\mathcal{S}(i)} \text{sim}(x_i, x_k)} = \frac{\sum_{j\in\mathcal{S}(i)} \text{SC}(x_j) \cdot y_j}{\sum_{k\in\mathcal{S}(i)} \text{SC}(x_k)} \tag{3}$$

where $\mathcal{S}(i) = \{j \mid j \in \mathcal{G}(i) \wedge x_j \in X_R\}$ is the set of nearby retained token indexes. Intuitively, the transformation result of a token should be more alike to that of a token with a higher similarity. This process involves $O(w^2)$ pairwise similarities per token, resulting in overall complexity of $O(w^2 N)$.

To improve the efficiency, we replace $\text{sim}(x_i, x_j)$ by $\text{SC}(x_j)$ to make reconstructions based more on the retained tokens with higher spatial coherence. However, this can result in the same transformation result of a skipped token within each grid, because of the weighted average of the transformation results of retained tokens. To relieve this issue, we restrict aggregation within a smaller areas by dividing each grid into smaller sub-grids with size $w_s \times w_s$ (e.g., $4 \times 4$ or $3 \times 3$) to leverage the spatial locality of the image, which effectively reduces the complexity to $O(N)$.

### 3.2.3 MICRO DESIGNS

**Alternating Grid Size.** Additionally, we observe visual artifacts at grid border (as shown in Fig. 3b) due to lack of inter-grid information exchange. We mitigate this issue by alternating grid sizes across transformer blocks. Specifically, for $32 \times 32$ tokens, we alternate grid size of 16 and 9. This strategy enables cross-border information flow over consecutive blocks, eliminating visible discontinuities.

$m$**-stride Token Retention.** In some scenarios (e.g., large chunk of background), all tokens in a sub-grid might be skipped, which leads to no retained tokens as reconstruction reference. To address this issue, we enforce always retaining a token for every $m$ tokens ($m \leq w_s$). Specifically, for transformer block with index $l$, we enforce the tokens at position $[i, j]$ with $(i + j - l) \mod m = 0$ to be retained, by subtracting a large offset on their spatial coherence scores, e.g., $m = 3$ in Fig. 3c.

### 3.3 PRUNING RATIO SCHEDULE

#### 3.3.1 MOTIVATION

The diffusion process maps noise to an image through sequential denoising steps. As shown in Fig. 4, early timesteps operate on representations dominated by noise and establish global semantics and coarse structure, while late timesteps refine fine-grained details such as textures and boundaries [6, 32]. Besides, within a timestep, transformer blocks exhibit different degrees of locality [5]. These observations imply that a uniform pruning ratio across timesteps and blocks is suboptimal.

Given that we prune tokens from pre-trained DiTs, we propose to determine the pruning ratio adaptively across blocks according to the statistics observed on real data. Specifically, we propose a progressive strategy during fine-tuning: we maintain a redundancy score for each block (e.g., the sum of spatial coherence for the next $\Delta k$ tokens to be pruned) and monotonically increase the pruning ratio for the block with the highest redundancy. This strategy leverages the empirical findings [31]: applying large pruning to a pretrained model can require substantial efforts for recovery, while progressive pruning leads to better results at comparable pruning levels.

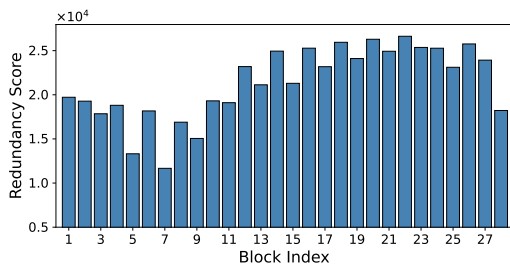 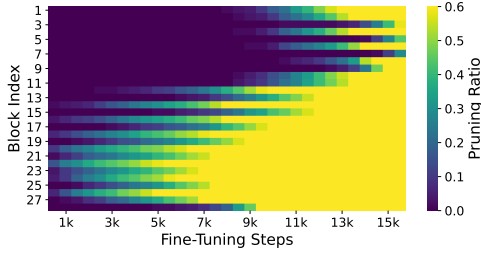

(a) Initial redundancy scores for the next $\Delta k$ tokens based on Equ.(4) across transformer blocks.

(b) Evolution of pruning ratios over fine-tuning, updated every T steps based on block-wise redundancy.

Figure 5: Visualization of coherence-guided progressive pruning ratios across blocks and fine-tuning steps for PixArt-$\alpha$-1024. The schedule will give more pruning to blocks with higher redundancy.

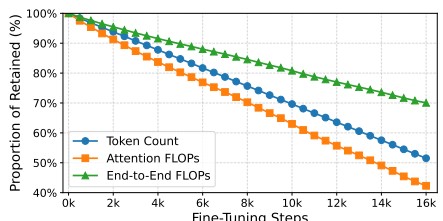 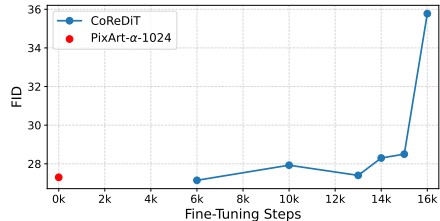

(a) Efficiency measured by the percentage of retained token count, attention FLOPs, and End-to-End FLOPs

(b) Quality assessed using the FID score. The red dot represents the pretrained model [8], while the blue line corresponds to the proposed CoReDiT method.

Figure 6: Metrics during progress pruning along the fine-tuning steps.

### 3.3.2 COHERENCE-BASED PROGRESSIVE PRUNING

Suppose that we prune $K_{l,s} = r(l,s) \cdot N$ tokens according to pruning ratio schedule $r(l,s)$ for block $l$ at denoising timestep $s$. During fine-tuning, we start from original block and gradually increase $K_{l,s}$ in fixed steps of $\Delta k$ every $T$ training iterations. To decide which block to increase pruning ratio next, we rank the spatial coherence score of input tokens $\{SC_i^l\}$ from highest to lowest, and define the redundancy score for top-$K$ tokens as $R_l(K) = \sum_{i=1}^{K} SC_i^l$. The redundancy of the next $\Delta k$ tokens in block $l$ is

$$\Delta R_l = R_l(K_{l,s} + \Delta k) - R_l(K_{l,s}) \qquad (4)$$

Fig. 5a presents the redundancy score for PixArt-$\alpha$-1024. Every $T$ training iterations, we greedily assign the next increment $\Delta k$ to the block with largest $\Delta R_l$, prioritizing blocks with the most redundant tokens at the current stage.

Additionally, pruning schedule should also reflect denoising phase: Early timesteps are noise-dominated and set global structure (more redundancy), while late timesteps refine local details (less redundancy). Accordingly, we adopt more aggressive pruning early and reduce it later. Since each distinct per-timestep ratio would instantiate a different computational graph, we adopt a two-phase schedule over denoising step $s$: $r(l,s) = \begin{cases} r(l), & 1 \leq s \leq 15 \\ c \cdot r(l), & 15 < s \leq 20 \end{cases}$ where $r(l)$ is the base pruning ratio for block $l$. Aligned with prior empirical observations [6, 32], this timestep-wise decay $c < 1$ captures phase-dependent redundancy while requiring only two computational graphs per block.

We demonstrate progressive pruning on PixArt-$\alpha$-1024 by incrementally increasing the pruned-token budget by $\Delta k = 64$ every $T = 15$ training steps, while capping the per-block pruning ratio at 60% of the 4096 tokens. Fig. 5a depicts substantial variation in block-level redundancy at initialization; as Fig. 5b indicates, the schedule adapts to these differences, directing more pruning to blocks with higher redundancy. For example, blocks 5 and 7 exhibit low redundancy scores and therefore maintain low pruning ratios throughout the fine-tuning. As shown in Fig. 6a, CoReDiT achieves significant efficiency gains by greatly reducing FLOPs within a short fine-tuning period, while maintaining quality as indicated by the FID in Fig. 6b.

| Model | FLOPs Reduction | | Image Quality | | |
|---|---|---|---|---|---|
| | Self-Attn | Total | FID ↓ | CLIP ↑ | IS ↑ |
| PixArt-$\alpha$-1024 | - | - | 27.3 | 31.6 | 37.77 |
| ToMeSD (25% ratio) [2] | - | -7% | 174.6 | 30.2 | 11.68 |
| DiffPruning [9] | - | -9% | 34.6 | 32.0 | - |
| EcoDiff [34] | - | -9% | 32.2 | 32.0 | - |
| DeepCache (N=2) [20] | - | -25% | 31.6 | 33.1 | 37.44 |
| CoReDiT ($r = 40\%$) | -48% | -24% | 28.7 | 32.1 | 37.96 |
| CoReDiT ($r = 40\%$) w/ distill | -48% | -24% | 27.4 | 31.9 | 36.85 |
| CoReDiT ($r = 45\%$) | -55% | -28% | 29.3 | 31.9 | 36.67 |
| CoReDiT ($r = 45\%$) w/ distill | -55% | -28% | 28.5 | 31.9 | 36.65 |

Table 1: Results on PixArt-$\alpha$-1024; $r$ denotes the average pruning ratio.

| Model | FLOPs Reduction | | Latency (Eff. Attn) | | Latency (Native Attn) | |
|---|---|---|---|---|---|---|
| | Self-Attn | Total | Self-Attn | Total | Self-Attn | Total |
| PixArt-$\alpha$-1024 | - | - | 0.68s | 1.68s | 2.16s | 3.17s |
| CoReDiT ($r = 45\%$) | -55% | -28% | 0.50s (-26%) | 1.50s (-11%) | 1.36s (-37%) | 2.38s (-25%) |

Table 2: Comparison of FLOPs and latency on Nvidia H100 GPUs, with batch size = 64.

## 4 RESULTS

### 4.1 EXPERIMENTAL SETUP

**Models and Training Datasets.** For text-to-image generation, we apply our approach to PixArt-$\alpha$-1024 [8] and PixArt-$\Sigma$-2048 [7]. We begin with the official checkpoints and apply our progressive pruning during fine-tuning on a 10K synthetic dataset as used in CLEAR [18], with images generated by FLUX.1-dev [14]. For video generation, we apply our method to MagicDrive-V2 [10] and fine-tune the released third-stage checkpoint on nuScenes [3].

**Progressive Pruning Configurations.** For text-to-image, We fine-tune PixArt-$\alpha$-1024 with a batch size of 20 on a single Nvidia H100 GPU and PixArt-$\Sigma$-2048 with a total batch size of 40 for across 8 Nvidia H100 GPUs. For progressive pruning, every 15 training steps, we select a transformer block and prune additional $\Delta k = 64$ tokens, with a per-block pruning ratio upper bound of 60% and timestep-wise pruning ratio decay $c = 0.25$. Besides, following [18], we apply distillation during fine-tuning for both model output $L_{pred}$ and the recent pruned block $L_{attn}$ based on loss $L_{distill} = L_{ori} + \alpha L_{pred} + \beta L_{attn}$, using the same hyperparameters $\alpha = 0.5$ and $\beta = 0.5$. For video generation, we target on spatial transformer blocks where the number of tokens is significantly greater than these of temporal blocks. We fine-tune the third-stage checkpoint with SP size of 8 over 8 Nvidia H100 GPUs. We limit the per-block pruning ratio $r(l)$ up to 50%, and adjust the pruning ratio according to denoising step $s$ for last 4 denoising steps with pruning ratio decay $c = 0.25$.

**Evaluation Metrics.** Following related work [18], we conduct evaluations on 10k caption-image pairs randomly sampled from MSCOCO 2014 validation dataset [17] with quality metrics: FID [11], CLIP text similarity [22], and Inception Score (IS) [26]. For video generation, we report FVD [28], LPIPS [33], PSNR, and SSIM [29] for video quality; we also evaluate mAP and mIoU by BEV-Former [15] for condition-video alignment.

### 4.2 MAIN RESULTS: PIXART-$\alpha$-1024

**Quality Metrics.** Table 1 compares baseline PixArt-$\alpha$-1024 with token-reduction methods. Compared with other existing work [2, 9, 34, 20], CoReDiT prunes 40% of tokens while reducing self-attention FLOPs by 48% and total FLOPs by 24%, and maintains quality close to the baseline: without distillation, FID rises modestly (27.3 vs 28.7), and both CLIP and IS improve. Adding distillation recovers gap of FID (27.4) with small drops in IS (36.85), demonstrating minimal perceptual or semantic loss. Fig. 7 qualitatively compare the generated images between PixArt-$\alpha$-1024 and CoReDiT (40%), showing that CoReDiT produces high-quality results: global semantics are preserved and no additional artifacts are introduced.

**Efficiency.** Table 2 reports end-to-end and self-attention latencies on Nvidia H100 GPUs under two attention backends: highly-optimized kernels XFORMERS.MEMORY_EFFICIENT_ATTENTION (Eff. Attn) and PyTorch native attention implementations (Native Attn). As shown, CoReDiT (45%

(a) Pretrained PixArt-$\alpha$-1024 model [8]

(b) Proposed CoReDiT with 40% FLOPs Pruning. The semantics are preserved and no additional artifacts.

Figure 7: Qualitative comparison to PixArt-$\alpha$-1024.

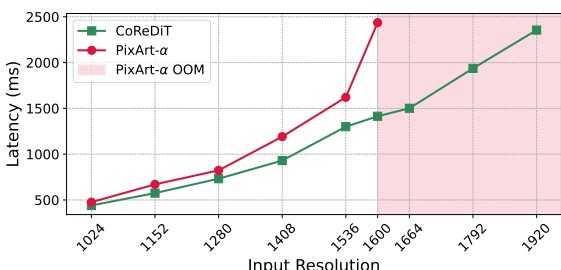

Figure 8: Per-block latency comparison on Qualcomm Snapdragon 8 Elite NPUs. OOM: out-of-memory.

| Model (ratio) | FLOPs Reduction | | Latency | | Image Quality | | |
|---|---|---|---|---|---|---|---|
| | Self-Attn | Total | Self-Attn | Total | FID ↓ | CLIP ↑ | IS ↑ |
| PixArt-$\Sigma$-2048 | - | - | 4.62s | 6.68s | 26.0 | 31.4 | 37.49 |
| CoReDiT ($r = 23\%$) w/ distill | -32% | -25% | 3.61s (-22%) | 5.67s (-15%) | 28.0 | 31.4 | 36.52 |

Table 3: Results on PixArt-$\Sigma$-2048, with latency measured on Nvidia H100 GPUs using Eff. Attention.

tokens pruned) leads to 26% self-attention latency and 11% end-to-end latency improvement with the memory-efficient backend. Under the native backend, the gains track FLOPs more closely: 37% for self-attention and 25% for end-to-end latency. The smaller percentage speedup reflects highly optimized kernels that do not scale quadratically with token count, while the native path exposes more of the quadratic attention savings, resulting in larger relative improvements.

Additionally, Fig. 8 depicts the on-device latency improvement for a single block in PixArt-$\alpha$, measured on Qualcomm Snapdragon 8 Elite NPUs. As can be seen, CoReDiT with 50% pruning ratio achieves up to $1.72\times$ (i.e., at 1600-resolution input) latency speedup. Notably, the original PixArt-$\alpha$ model encounters out-of-memory (OOM) error when the input resolution exceeds 1600, while CoReDiT supports resolutions up to 1920, with compatible latency to that of PixArt-$\alpha$ at 1600 resolution. Overall, the results indicate that substantial compute and runtime savings can be achieved with minimal perceptual and semantic degradation, showing favorable efficiency–quality trade-offs.

## 4.3 HIGH RESOLUTION: PIXART-$\Sigma$-2048

We apply CoReDiT to PixArt-$\Sigma$-2048 to evaluate the performance on high-resolution image generation. As shown in Table 3, at an average pruning level of 23%, CoReDiT reduces self-attention FLOPs by 32% and total FLOPs by 25%, translating into a 22% reduction in self-attention latency and a 15% end-to-end latency reduction at batch size 32 with xformers memory-efficient attention. Besides, CoReDiT retains high image quality: CLIP remains the same, while FID and IS show small drop (26.0 v.s. 28.0 and 37.49 v.s. 36.52, respectively), likely due to the resolution mismatch, i.e., our fine-tuning dataset is at 1K (upscaled to 2K) while evaluation is conducted against native 2K images. This mismatch can affect quality, since the upscaled 1K data lacks the high-frequency details and long-range structures present in 2K images. We expect that fine-tuning on 2K data would narrow this gap and achieve better FLOPs savings into quality-preserving speedups.

| Model | FLOPs Reduction | | Video Quality | | | | Cond. Alignment | |
|---|---|---|---|---|---|---|---|---|
| | Self-Attn | Total | PSNR ↑ | LPIPS ↓ | SSIM ↑ | FVD ↓ | mAP ↑ | mIoU ↑ |
| MagicDrive-V2 | - | - | 14.28 | 0.422 | 0.372 | 107.8 | 18.4% | 21.5% |
| CoReDiT ($r = 26\%$) | -39% | -8% | 14.25 | 0.424 | 0.378 | 119.8 | 18.1% | 21.2% |

Table 4: Video generation results on video quality and conditional alignment via 3D object detection

| Experiment | Image Quality | | |
|---|---|---|---|
| | FID ↓ | CLIP ↑ | IS ↑ |
| PixArt-$\alpha$-1024 | 27.3 | 31.6 | 37.77 |
| CoReDiT ($r = 45\%$) | 28.5 | 31.9 | 36.65 |
| Random selection ($r = 23\%$) | 644.5 | 21.5 | 1.00 |
| Disable reconstruction ($r = 30\%$) | 30.4 | 31.7 | 34.20 |
| Uniform ratio ($r = 41\%$) | 32.3 | 31.5 | 33.68 |

Table 5: Ablation study on PixArt-$\alpha$-1024, with distillation in all experiments.

## 4.4 VIDEO GENERATION: MAGICDRIVE-V2

Recent trends in video generation research emphasize higher resolutions and longer sequences, which lead to increased token sequences and substantial growth in computational demands. As a result, architectures like DiT have become essential. However, This also underscore the urgency of reducing unnecessary computation. Therefore, we extend CoReDiT, which is broadly applicable across vision tasks and agnostic to input modalities, to the state-of-the-art video generation DiT for autonomous driving - MagicDrive-V2 [10]. Our integration demonstrates that redundancy in the model can be reduced. Specifically, we achieve a 39% FLOPs reduction in self-attention within the spatial transformer blocks, while preserving visual quality. Quantitative metrics such as PSNR, LPIPS, and SSIM confirm that our method maintains per-frame perceptual quality. Additionally, mAP and mIoU also suggest that our method provides stable conditional alignment.

Currently, CoReDiT is applied only to spatial transformer blocks in which each frame is processed independently. Consequently, we observed a drop in FVD, which reflects temporal consistency in the generated video. Future work will focus on integrating temporal consistency objectives, such as cross-frame coherence, to mitigate this limitation. We expect such enhancements will improve FVD performance while maintaining the computational efficiency demonstrated by our current approach.

## 4.5 ABLATION STUDY

Table 5 presents our ablation studies to evaluate the impact of key design choices in our approach: (1) *Random Token Selection.* To evaluate the importance of spatial-coherence based token selection, we conduct experiments of randomly sampling tokens to skip in each block; this yields extremely poor quality (FID 644.5 at 23% pruning), proving that spatial-coherence based selection is effective. (2) *Skipping Tokens without Reconstruction.* To illustrate the contribution of reconstruction, we disable it and simply skip selected tokens. At 30% pruning ratio, FID degrades to 30.4, indicating that reconstruction is necessary to preserve semantic fidelity. (3) *Uniform Pruning then Fine-tuning.* To justify the effectiveness of progressive pruning, we apply a uniform 50% per-block pruning to the pretrained model and then fine-tune; this results in worse FID 32.3 at average $40.6\%$ pruning, highlighting the advantage of progressive, block-adaptive pruning.

## 5 CONCLUSION

We propose CoReDiT, a token pruning framework for DiTs that exploits spatial coherence. CoReDiT combines a pre-attention spatial-coherence selector to bypass redundant tokens, a similarity-guided reconstruction operator to preserve locality and texture for skipped tokens, and a progressive, block-adaptive schedule that allocates pruning where redundancy is highest. On both image and video generation, CoReDiT results in substantial inference-time speedups while maintaining visual fidelity. Future work includes adaptive grid partitioning and cross-temporal coherence modeling for video generation.

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

## A    MORE EXAMPLES ON SPATIAL COHERENCE SCORE

Fig. 9 provides additional generations from PixArt-$\alpha$-1024 alongside their spatial-coherence maps. Across diverse prompts, large uniform regions (e.g., sky, roads) consistently exhibit high spatial coherence (indicating redundancy) while detail-rich structures (e.g., object boundaries, small parts) remain low spatial coherence. This pattern aligns with the behavior of our token selector: high-coherence areas are pruned more aggressively, whereas salient structures are preserved.

Fig. 10 presents the evolution of spatial coherence during inference for the image in Fig. 2b, shown across diffusion steps and blocks. Early steps display broadly elevated coherence (coarse structure), mid steps accentuate background redundancy, and late steps concentrate low-coherence pockets around edges and details. This progression motivates our block-adaptive schedule: we allocate larger pruning budgets where coherence remains high and throttle pruning where coherence drops, preventing over-pruning of emerging details and preserving final image fidelity.

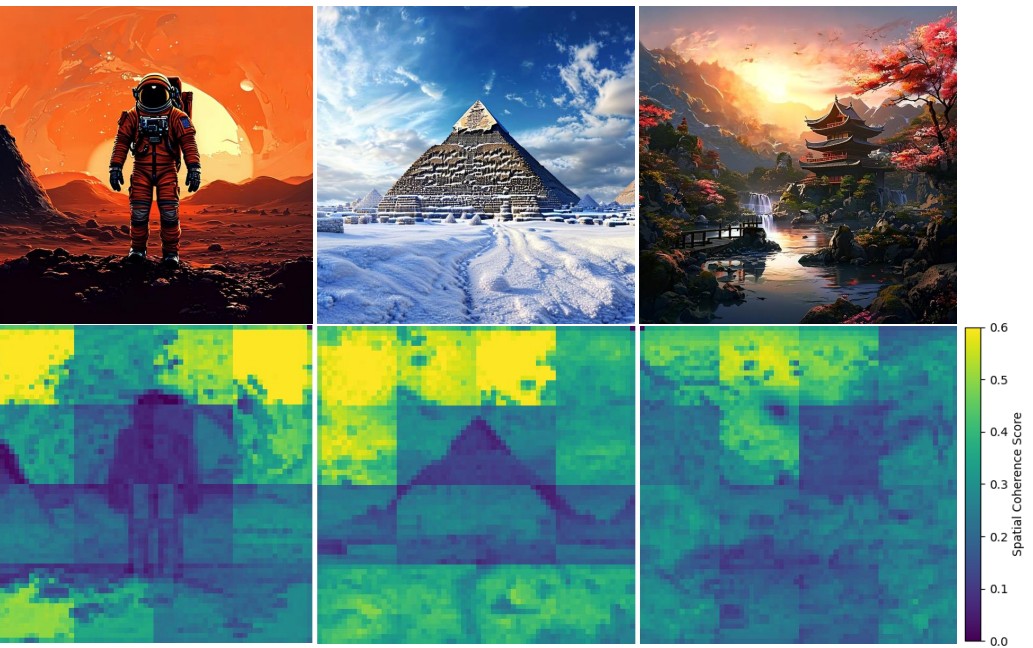

Figure 9: More examples on spatial coherence score.

## B    MAGICDRIVE-V2 FVD EVALUATION

We note a limitation of FVD evaluation: there is a substantial distribution shift from Kinetics-400 which is a human action recognition dataset (used to train the I3D [4] feature extractor) to nuScenes which is a driving dataset (used to fine-tuning MagicDrive-V2). This mismatch leads to suboptimal feature representations when using I3D, causing the FVD metric to mischaracterize the perceptual and semantic quality of generated driving videos. In particular, we find that the FVD changes drastically during fine-tuning, suggesting that the metric is highly sensitive in representation space that may not correspond to meaningful improvements in visual quality.

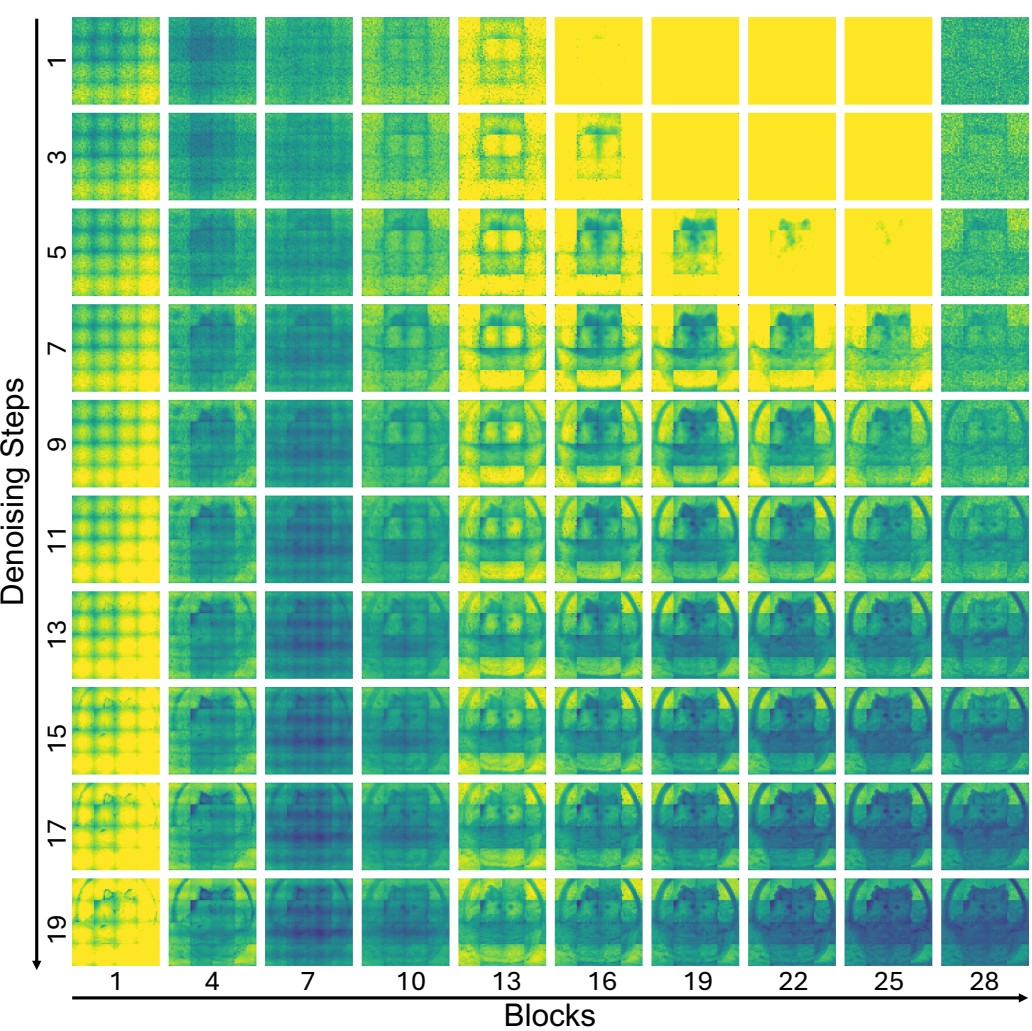

Figure 10: Full visualization of spatial coherence score (brighter = higher) during model inference for generating the image in Fig. 2b.

