# OpenReview forum: "COREDIT: SPATIAL COHERENCE-GUIDED TOKEN PRUNING AND RECONSTRUCTION FOR EFFICIENT DIF- FUSION TRANSFORMERS"
_ICLR.cc/2026/Conference — ICLR 2026 Conference Withdrawn Submission_

### Official Review · Reviewer_ZgC6 · 2025-10-29

**Soundness:** 2
**Presentation:** 2
**Contribution:** 2
**Rating:** 2
**Confidence:** 5

**Summary:**

The paper studies the problem of token pruning for diffusion transformer models. The paper proposes CoReDiT, which first utilizes spatial coherence among tokens to estimate the redundancy and partitions the tokens into a skip set and a retain set. After self-attention computation on the retain set, to match the original token shape, the method reconstructs the skip set tokens using a strategy that is similar to attention by averaging outputs of the retain set based on their similarities to the skip set. CoReDiT also fin-tunes the reduction rate of various blocks to find the best redundancy-quality trade-off. The paper shows that CoReDiT can achieve FLOPs reductions and speedups using the Pix-art-1024 model.

**Strengths:**

1. The paper identifies the problem of applying the token pruning method in prediction models directly to the generation models, and offers an intuitive way to do the adaptation.

2. The proposed method can achieve wall-clock-time speedups.

**Weaknesses:**

1. The proposed method can achieve a reasonable FLOPs reduction. However, its wall-clock-time speed-up is not quite significant, meaning that the overhead of the proposed method could be relatively costly. Moreover, the paper does not give a comprehensive analysis of the overhead, i.e., what its overall theoretical complexity is, the FLOPs, and the wall-clock-time costs.

2. The proposed method only accelerates the attention part of the network using token pruning. Compared to other acceleration methods, such as sparse attention, the benefits of the token pruning strategy are that it can also be applied to the feed-forward part of the network. On image sizes of 1024, their costs are comparable. Moreover, as the proposed method utilizes a method that is similar to attention to reconstruct the skip tokens, the overall strategy is quite similar to sparse attention. Therefore, I think comparison with sparse attention methods should also be included.

3. Discussions and relations to token merge approaches are not well addressed. ToMe is briefly introduced but as a pruning strategy. Other merge strategies, such as ToFu, ToMA, and ToDo, are not discussed.

4. The proposed method requires finetuning to find the optimal redundancy reduction for different blocks, while many other methods are plug-and-play.

5. More qualitative results should be provided. Moreover, though semantically similar, the generated images of the proposed method differ from the original model, while other methods, such as ToMe, typically preserve the overall image structure.

**Questions:**

Please address the weaknesses.

---

### Official Review · Reviewer_d7Kg · 2025-11-01

**Soundness:** 3
**Presentation:** 3
**Contribution:** 2
**Rating:** 4
**Confidence:** 4

**Summary:**

This paper presents CoReDiT, a spatial coherence guided token pruning and reconstruction framework for Diffusion Transformers (DiTs). The method prunes redundant tokens in locally smooth regions and reconstructs their outputs using similarity-weighted aggregation from nearby retained tokens. A progressive, block-adaptive pruning schedule further adjusts pruning ratios across layers and diffusion steps based on redundancy statistics.

Applied to PixArt-α/Σ and MagicDrive-V2, CoReDiT achieves up to 55% self-attention FLOPs reduction and 1.3~1.7x speedups on GPUs/NPUs, while maintaining close-to-baseline image and video quality. Ablations confirm that spatial coherence is an effective redundancy signal and that reconstruction is key for preserving fidelity.

Overall, CoReDiT offers a practical approach to efficiency in diffusion transformers, leveraging spatial coherence for structured pruning.

**Strengths:**

1. Clear and well-structured methodology. The paper clearly explains the spatial coherence based pruning, reconstruction, and progressive scheduling mechanisms, supported by intuitive figures and mathematical formulations.

2. Comprehensive empirical validation. Experiments on both image (PixArt-α/Σ) and video (MagicDrive-V2) diffusion models convincingly demonstrate effectiveness, showing consistent FLOPs reduction and runtime speedups.

3. Practical significance. CoReDiT provides a general, architecture-agnostic token pruning method that preserves fidelity while enabling high-resolution generation on resource-limited devices, making it practically valuable.

**Weaknesses:**

1. Heuristic design with heavy manual tuning. The method’s use of attention or feature similarity for token importance is intuitive but not sufficiently novel, closely resembling prior token-merging approaches that also perform merge/unmerge operations analogous to pruning/reconstruction here. Moreover, it depends on multiple hand-tuned hyperparameters, such as grid size, pruning ratio schedules, and coherence thresholds, making the approach less principled, sensitive to tuning, and harder to generalize across models and datasets.

2. Unclear computation overhead and locality bias. Computing spatial coherence via cosine similarity could be expensive; the paper does not report its runtime cost relative to total inference time. Moreover, restricting similarity computation within local grids effectively enforces a local attention pattern whose optimality is not justified.

3. Unstable fine-tuning behavior. Figure 6(b) shows FID increase with longer fine-tuning, suggesting instability in the progressive pruning schedule or reconstruction strategy. The cause of this performance drop is not well analyzed.

4. Limited scope of efficiency optimization. The method only prunes tokens in the attention layers, leaving MLP/FFN layers untouched, which is often dominant in overall FLOPs or latency, thus limiting real-world efficiency gains.

5. Missing comparisons with strong learning-based baselines. Although some related works are mentioned, the learning-based efficiency methods such as Learning to Cache (https://arxiv.org/abs/2406.01733), DynamicDiT (also used spatial information; https://arxiv.org/abs/2410.03456), DiffCR (https://arxiv.org/abs/2412.16822), and ToCA (https://arxiv.org/abs/2410.05317) are not included in the experimental comparison. These models represent more adaptive and principled approaches to efficiency, and their absence makes it difficult to contextualize CoReDiT’s effectiveness and trade-offs relative to current state-of-the-art adaptive pruning or caching techniques.

**Questions:**

See weaknesses.

---

### Official Review · Reviewer_83zQ · 2025-11-02

**Soundness:** 3
**Presentation:** 3
**Contribution:** 2
**Rating:** 4
**Confidence:** 4

**Summary:**

This paper introduces COREDiT, a token pruning and reconstruction framework designed to improve the computational efficiency of Diffusion Transformers (DiTs) in image and video generation tasks. The central idea leverages a spatial coherence score to estimate token redundancy within local grids, selectively skipping computation for high-coherence tokens, and reconstructing their outputs via similarity-weighted aggregation from retained tokens. An adaptive, progressive pruning schedule further tunes the pruning ratio across blocks and denoising steps based on redundancy statistics. The method reports up to 55% self-attention FLOPs reduction and clear runtime speedups on GPUs and mobile NPUs across multiple state-of-the-art DiT backbones, while maintaining strong visual quality and supporting higher resolution synthesis.

**Strengths:**

1. This paper offers a coherent, well-motivated framework addressing token redundancy by quantifying local spatial coherence, enabling hardware-friendly and effective token selection.
2. The computation of spatial coherence and its direct connection to token selection is clearly presented.

**Weaknesses:**

1. How does CoReDiT differ, both methodologically and empirically, from recent cluster-driven or caching-based approaches (e.g., ClusCa, TokenCache, CAT Pruning, DyDiT)?

2. Can the authors provide a more rigorous empirical or theoretical justification for the use of spatial coherence scores in place of similarity in reconstructing the skipped tokens? Are there failure cases or specific patterns where this heuristic degrades quality, and how sensitive is the approach to $w_s$?

3. What is the quantitative effect of alternating grid sizes, sub-grid sizes $w_s$, and stride retention on both quality (FID, CLIP, IS) and speedup? Are there optimal choices, or is the method robust to these design decisions?

4. Do the authors plan to evaluate the impact of their scheme on temporal consistency in video generation, beyond spatial quality? Could they integrate a cross-frame coherence loss or show frame-level FVD breakdowns in MagicDriveV2?

5. Would re-finetuning the high-resolution PixArt-$\Sigma$-2048 using evaluation-scale 2K images close the observed FID gap?

6. Can the authors supplement ablations by analyzing the progressive pruning schedule hyperparameters ($\Delta k$, $T$, $c$)? Additionally, could the authors provide more analysis on the block-wise redundancy evolution over time (e.g., does adaptivity always out-perform uniform schedules?)

7. Have the authors tested their approach relative to recent training-free efficiency schemes or compared with fully unstructured pruning/quantization for DiTs? What is the added value of CoReDiT’s fine-tuning step versus, e.g., ToMA or attention-driven methods on deployment metrics?

**Questions:**

see Weaknesses

---

### Note · Authors · 2025-12-03

I have read and agree with the venue's withdrawal policy on behalf of myself and my co-authors.